# Development and Biological Characterization of Cancer Biomimetic Membrane Nanovesicles for Enhancing Therapy Efficacy in Human Glioblastoma Cells

**DOI:** 10.3390/nano14221779

**Published:** 2024-11-05

**Authors:** Martina Massarotti, Paola Corna, Aromita Mallik, Gloria Milanesi, Claudio Casali, Lorenzo Magrassi, Sergio Comincini

**Affiliations:** 1Department of Biology and Biotechnology, University of Pavia, 27100 Pavia, Italy; martina.massarotti01@universitadipavia.it (M.M.); aromita.mallik01@universitadipavia.it (A.M.); gloria.milanesi@unipv.it (G.M.); claudio.casali@unipv.it (C.C.); 2Department of Clinical Surgical Diagnostic and Pediatric Sciences, University of Pavia, 27100 Pavia, Italy; paola.corna@unipv.it (P.C.); lorenzo.magrassi@unipv.it (L.M.)

**Keywords:** membrane-camouflaged nanovesicles, glial tumors, Rose Bengal anticancer

## Abstract

As nanocarriers of a new generation, biomimetic nanovesicles are an emerging class of therapeutic tools whose surface is integrated or fabricated with biomaterials capable of mimicking the biological features and functions of native cells. Thanks to this, biomimetic nanovesicles, in particular, those made by plasma membrane moieties, possess greatly improved biocompatibility, high target specificity, a long retention time, and minimal undesired immune responses. For these reasons, a multitude of progenitor cells including cancer ones were employed as templates to generate biomimetic or membrane-camouflaged nanovesicles hosting different therapeutic compounds. In this contribution, different membrane-derived biomimetic vesicles (M-NVs) were generated by osmotic lysis or plasma membrane isolation approaches from normal and cancer cell lines and assayed against in vitro models of human glioblastoma. M-NVs were compared in their cellular internalization degrees of DNA and proteins, morphologically and molecularly characterized, expressing an extracellular membrane-associated marker. Then, Rose Bengal (RB), a photoactivable drug characterized by a relatively low cellular uptake, was incorporated into nascent glioblastoma-derived M-NVs and finally administered to homotypic receiving cells, showing an increased degree of internalization as well as induced cytotoxic effects, even in the absence of photodynamic direct stimulation. Similar results were also obtained assaying lyophilized M-NVs loaded with RB. In conclusion, M-NVs generated by cell membranes effectively deliver several cargoes, including therapeutic molecules, maintain functionality after lyophilization, and show significant internalization effects, making them a promising strategy for therapeutic applications against human glioblastoma cells.

## 1. Introduction

Extracellular vesicles (EVs) are biological units surrounded by plasma membranes that are produced and released by cells in physiological and/or pathological conditions. EVs are classified into three main categories according to their size and origin: exosomes, produced by the endosomal pathway (diameter range: 30–140 nm), microvesicles, derived from the budding process of plasma membranes (200–1000 nm), and apoptotic bodies, obtained as a byproduct of apoptotic cell death (50–5000 nm) [1,2]. The biochemical signature of EV membrane protein and lipid composition and their cargo content are highly heterogeneous due to different cellular origins as well as varying downstream biological effects in recipient cells. Generally, EV membranes are defined by a lipid bilayer with extracellular exposed proteins like tetraspanins, integrins, and immunomodulatory ones [3] contributing to crucial roles in cells communication. EVs were largely assayed for the delivery of different cargo contents (i.e., RNAs, proteins, drugs) into target cells. How-ever, due to the intrinsic membrane-enclosed architecture of the EVs, the capability to reach efficient loading yields using approaches like sonication, electroporation, freeze-thaw cycles, or cell transfection protocols is still limited [4].

Previously, nanoparticle-based drug delivery systems have been extensively investigated in different pathological conditions [5]. Primarily, synthetic nanoparticles (NPs) were largely produced and assayed for their high drug loading capacity and various molecular and chemical modifications [6]. However, despite in vitro and in vivo functionality, NPs did not display a complete therapeutic effect in clinical settings, mainly due to unspecific clearance, reduced biocompatibility, and off-target toxicity [7]. Another class of cargo vehicles, Liposomes, as assembled phospholipid bilayers, have been largely assayed as drug carriers [8]. However, despite preclinical evidence, in vivo instability due to the lack of a complete membrane architecture remains a major limitation [9]. These biological limitations resulted in the development of ingenious approaches to prepare cell-membrane-based coated nanoparticles as “Trojan horses” to evade immune responses and extend their circulation time, while simultaneously ensuring a high drug loading content [10,11]. Currently, several biomimetic systems can be produced by combining different cell membrane templates with different nanomaterials and drug combinations [12]. Therefore, cell membranes were adopted as an alternative template for developing a novel class of biological vectors referred to as biomimetic or membrane-camouflaged nanovesicles. These can be designed as suitable drug delivery systems by incorporating active compounds or genetic material [13]. The rationale is that these delivery systems may mimic natural cellular communication processes better and deliver therapeutic drugs to desired targets with an increased specificity and stability within an organism [14]. 

Biomimetic membrane-derived nanovescicles (M-NVs) originated by using cell lysis methods, removing the intracellular content, and isolating the membrane-associated components. As originally described, M-NVs were mainly derived from platelets [15], leukocytes [16], mesenchymal stem cells [17], and cancer cells [18]. 

In vivo studies demonstrated the relatively high stability of M-NVs in biological fluids as well as the ability of these therapeutic or theranostic vectors to cross different biological barriers like the blood–brain barrier (BBB). Furthermore, the presence of the CD47 signal regulatory protein component, also referred to as the “do not eat me” pathway, in nascent M-NVs can confer a favorable clearance protection of M-NVs from macrophages, subsequently extending their lifespan and promoting proper therapeutic targeting [13,14]. 

Glioblastoma (GBM), categorized as a grade IV astrocytoma, is the most prevalent, aggressive, and lethal primary brain tumor in adults. Despite continuous advances in neurosurgery, chemotherapy, and radiotherapy practices, GBM remains one of the most treatment-resistant malignancies and its relapse is almost always inevitable [19]. The conventional treatment of GBM consists primarily of surgical resection, followed by a radiotherapy combined scheme in the presence of the alkylating agent temozolomide (TMZ) [20].

However, due to difficulty releasing drugs into the central nervous system in therapeutic amounts and the associated side effects, direct therapeutic administration schemes show relatively low efficiency [21,22,23]. Hence, significant effort was devoted towards encapsulating drugs into a broad category of shuttling systems like polymeric NPs, such as dendrimers, polymer micelles, or nanospheres, coupled with silica, iron, gold, or graphene NPs, lipid-based NP biodegradable nanogels, carbon dots, and, of course, into biomimetic vesicles [24,25]. These systems were also further functionalized by including targeting ligands that selectively recognize specific or overexpressed receptors on tumoral cells (folate, transferrin, neurokinin-1, or v3 integrin receptors) [22]. Several synthetic nanocarriers were designed and characterized for use in in vitro and in vivo brain tumor experimental models to combat the main physiological and anatomical problems that contribute to the resistance of GBM to therapeutic treatments [26].

In this contribution, we provide a relatively simple procedure to generate M-NVs along with their functional evaluation in human GBM cells.

## 2. Materials and Methods

### 2.1. Cell Culture Conditions and Viability Assays

High-grade human astrocytoma (i.e., U138-MG, U373-MG) and human cervix cancer (HeLa) cell lines were obtained from the American Type Culture Collection (Manassas, VA, USA); human primary fibroblast cells were provided by Dr. Stefanini (IGM-CNR, Pavia, Italy). Cells were routinely grown as monolayers at 37 °C in high-glucose Dulbecco’s modified Eagle’s medium supplemented with 10% fetal bovine serum and 100 μg/mL of penicillin–streptomycin (both from Euroclone, Milan, Italy) under atmosphere controlled at 5% CO_2_. 

MTT viability assays were performed as already described [27]. Briefly, cells were seeded at a density of 4 × 10^3^ cells/well in 96-well plates in a volume of 200 μL for 24 h and then treated with cargo-less biomimetic vesicles. At 24 and 48 h post treatment (p.t.), 20 μL of Cell Titer 96 AQueous One Solution Reagent (Promega, Madison, WI, USA) was added in each well and incubated for 2 h at 37 °C. Then, absorbance was measured using a microplate reader (Sunrise, Tecan, Männedorf, Switzerland) at a wavelength of 492 nm. All experiments were performed in triplicate with independent assays.

### 2.2. Generation of Membrane Biomimetic Nanovesicles (M-NVs)

M-NVs were obtained following two protocols. For the first process, referred to as osmotic lysis, cells were seeded (10^5^ cells) in a 60 mm diameter dish layered by 500 μL polylysine (Sigma, St. Louis, MO, USA) to increase adhesion efficiency. As soon as the cells reached confluence (~95%), the medium was removed, and cells were washed with PBS and incubated for 12 h with hypotonic sterile water, which resulted in the release of cytosolic and nuclear components. Then, the polylysine-attached membranes were scraped and collected into 1 mL of PBS. They were further disrupted using a 1 mL syringe needle 10 times, vortexed, and centrifuged at 16,000× *g* for 5 min at room temperature (RT). Membrane-derived portions were then incubated for 1 h under agitation at RT to induce inter- or intra-molecular recircularization events, thus generating M-NVs that were then quantified and assayed in diameter ranges the Tunable Resistive Pulse Sensing (TRPS) principle (see Section 2.5). M-NVs were stored in PBS at −20 °C.

The Minute Plasma Membrane Protein Isolation and Cell Fractionation Kit (Invent Biotechnologies, Plymouth, MN, USA) was used for the second approach to generate M-NVs. After trypsinization of the same amount of cells as the previous method, cell pellets were incubated for 15 min on ice in 500 μL of buffer A adopted for whole cell lysis. Lysates were then vortexed (20 s), following which cell suspensions were transferred to filter cartridges and centrifuged at RT for 30 s at 16,000× *g*. The filters were discarded and the pellets were resuspended by vortexing for 10 s. The cell suspensions were centrifuged at 700× *g* for 1 min and pellets were obtained containing intact nuclei. The supernatants were then transferred to fresh 1.5 mL microcentrifuge tubes and centrifuged at 4 °C for 30 min at 16,000× *g*. The supernatants, containing cytosol fractions, were removed and pellets of total membrane fractions were obtained. As described above, M-NVs were incubated to induce inter- or intra-molecular recircularization, quantified, morphometric-measured by TRPS, and finally stored in PBS at −20 °C.

### 2.3. M-NV Fluorescent Staining and Molecular Cargoes

The lipophilic CellBrite Green dye (CBG, Clinisciences, Rome, Italy) was used to stain the plasma membranes of generated M-NVs. M-NVs (6 × 10^7^ particles, quantified using qNano Gold instrument, see Section 2.5) were incubated with 0.5 μL CBG dye for 15 min at RT under agitation. Then, fluorescent M-NVs were purified from unbound dyes using Exosome Purification Columns (Invitrogen, Carlsbad, CA, USA). To stain the mitochondria in living cells, MitoTracker Deep Red (Invitrogen) was employed. The cells were incubated with 0.1 μM dye at 37 °C for 15 min and then visualized by Nikon Eclipse TS100 (Shinagawa City, Japan) inverted fluorescent microscope using oil immersion objective at 100×.

Different molecules were assayed for their capability to be introduced into M-NVs (6 × 10^7^ particles). These molecules were generated by the Minute Plasma Membrane Protein Isolation and Cell Fractionation Kit (Invent Biotechnologies, Plymouth, MN, USA) using U138-MG cells as templates. As nucleic acid cargo, the expression plasmid pEGFP-N1 (1 μg) [28] was electroporated by Neon NxT (Invitrogen, ΔV = 600 V, pulse width = 10 ms, pulse number = 20), transfected by Lipofectamine 2000 reagent (5 µL, Invitrogen), or administered as M-NVs (pEGFP) into U138-MG cells (105 in 30 mm diameter dishes). The expression of EGFP was monitored using an inverted fluorescent microscope after 48 h at 40×.

As a protein cargo, M-NVs were loaded with an anti-CD47 primary antibody (0.5 µg, Sony Biotechnology, #2215510, San Jose, CA, USA) fluorescently conjugated using the Rapid DyLight450 labeling kit (Bio-Rad, Hercules, CA, USA), as described in a previous study [28], and administered to recipient U138-MG cells.

As a drug cargo, Rose Bengal sodium salt (RB, >95% purity, Sigma) was resuspended in sterile water, filtered by 0.22 µm pores, and administered at 12.5 μM for different time intervals to U138-MG cells. RB was also incorporated into U138-MG-derived M-NVs and administered to the same cells. Light irradiation for RB was performed using the Chemidoc MP instrument (Bio-Rad) 48 h post treatment and setting excitation/emission to 562/576 nm with intensity = 2.76 mW/mm^2^, as described previously [29].

### 2.4. Imagestream Flow Cytometry Analysis

M-NVs or cell samples were analyzed using the Amnis ImageStreamX MarkII instrument (Luminex, ThemoFisher Scientific, Waltham, MA, USA) with the following laser settings (100 mW 488 nm, 70 mW 785 nm), as previously documented [30]. The ISX objective 60× (NA = 0.9; DOF = 2.5 µm, core size = 7 µm) was adopted to visualize the cell internalization of CBG-stained M-NVs. To analyze CBG-stained M-NVs alone, a “High gain mode” was selected with a 60× objective. CBG signals were collected in channel 2 (480–560 nm filter), while channel 6 (745–800 nm filter) was used for scatterplot (SSC) detection. Data analysis was performed using Amnis IDEAS software (Amnis, v 6.2).

### 2.5. M-NV Size Distribution and Concentration Analysis

The qNano Gold instrument (Izon Science, Christchurch, New Zealand) was employed to measure the size distribution and concentration of the isolated M-NVs using the Tunable Resistive Pulse Sensing (TRPS) principle [31]. Briefly, 35 μL of M-NVs were analyzed using an NP250 Nanopore (Izon Science) by applying 49 mm stretch, 0.34 V, and 10 mBar parametric conditions. The calibration particles (CPC100, Izon Science) were assayed before the experimental samples under identical conditions. Size and concentration (2000 events each) values of M-NVs were finally determined using the Izon Control Suite version 3.1 as already described [32].

### 2.6. Transmission Electron Microscopy Examinations of M-NVs

M-NVs were visualized by transmission electron microscopy (TEM), as already reported [33]. In detail, 20 μL drops of the isolated M-NVs in D-PBS were placed on a Parafilm (Sigma) sheet, and a 300-mesh nickel grid (covered with a Formvar-carbon film) was floated onto the drops, allowing it to stay for 2 min. The grids were then rapidly blotted with filter paper and negatively stained with a 3% uranyl acetate solution for 2 min, blotted on paper, and observed directly on a JEM 1200 EX II (JEOL, Peabody, MA, USA) electron microscope equipped with a MegaView G2 CCD camera (Olympus OSIS, Tokyo, Japan) and operating at 120 kV.

### 2.7. Protein Immunoblotting Analysis of M-NVs

The evaluation of specific proteins in U138-MG-plasma-membrane-generated M-NVs was performed by isolating M-NVs from 5 × 10^6^ U138-MG cells (grown in 10 cm diameter plates) using the Minute Plasma Membrane Protein Isolation and Cell Fractionation Kit as previously described. After plasma membrane isolation, total protein content was determined by Qubit fluorimeter, using the Protein Assay kit (Invitrogen, Waltham, MA, USA) following the manufacturer’s instructions. Before loading in SDS-PAGE gel, M-NVs extracts (40 µg) were boiled in Laemmli sample buffer (2% SDS, 6% glycerol, 150 mM B-mercaptoethanol, 0.02% bromophenol blue, and 62.5 mM Tris-HCl pH 6.8). After electrophoresis, M-NVs were transferred onto a nitrocellulose membrane Hybond-C Extra (GE Healthcare, Milan, Italy). Membranes were blocked with 5% nonfat milk in PBS containing 0.1% Tween 20 (*v*/*v*) and incubated overnight at 4 °C with primary antibodies. The primary antibodies (diluted 1:2000) were anti-Annexin V (Cell Signaling, Danvers, MA, USA, #8555), anti-CD47 (Sony Biotechnology, #2215510), anti-CD63 (Cell Signaling, #52090), anti-CD81 (Cell Signaling, #56039), and anti-HSP 70 (Cell Signaling, #4872). Species-specific peroxidase-labeled ECL secondary antibodies (Cell Signaling, diluted 1:4000) were employed. Protein signals were revealed by the Weststar Hypernova Kit (Cyanagen, Bologna, Italy) and visualized using the Chemidoc MP system (Bio-Rad).

### 2.8. M-NV Lyophilization

Frozen M-NVs derived from U138-MG (6 × 10^7^ particles resuspended in 100 µl sterile 0.22 µm filtered water) were exposed to a 30 min dehydration cycle at 45 °C using the Eppendorf Concentrator 5301 instrument (Hamburg, Germany). Lyophilized M-NVs were then stored at −20 °C and solubilized with 100 µl sterile 0.22 µm filtered water before use.

### 2.9. Clonogenic Assay of RB-Loaded M-NVs

To evaluate the long-term effect of RB-loaded M-NVs, referred to as M-NVs (RB), a clonogenic survival assay was performed according to previously documented protocol [27]. Briefly, following 48 h of incubation, 1000 cells previously treated with RB (12.5 μM) or with lyophilized M-NVs (RB) at the same drug concentration were transferred onto 60 mm diameter plates in a standard high-glucose medium for one week. The cells were then fixed with ethanol and stained with Crystal Violet (0.5%, *v*/*v*). ImageJ software (v. 1.1.4) colony-counter (https://imagej.nih.gov/ij/plugins/colony-counter.html, accessed on 29 January 2024) was used for colony calculations.

## 3. Results

### 3.1. Generation of Plasma-Membrane-Derived Nanovesicles (M-NVs)

For evaluating suitable approaches to generate plasma-membrane-derived nanovesicles (M-NVs), two methods were considered. The first approach was based upon the adhesion of cells to a polylysine layer, followed by hypotonic lysis with distilled water that released cytoplasmic and nuclear contents and trapped plasma membranes, which were recovered by scraping [34]. Hence, U138-MG cells were seeded (10^5^ cells) in a 60 mm diameter dish covered by a polylysine layer (500 μL). When cells were almost fully confluent (~95%), the medium was removed, and cells were washed with PBS and incubated for 12 h in sterile water (Appendix A). Plasma membrane fractions were then collected by scraping, mechanically fragmented by 1 mL syringe needle, vortexed, and reassembled onto spherical structures. These structures were stained with CBG plasma membrane dye and analyzed for concentration and diameter sizes by qNano Gold instrument. 

The latter approach was based on the direct biochemical isolation of plasma membranes, adopting the Minute Plasma Membrane Protein Isolation and Cell Fractionation kit, and therefore referred to the direct plasma membrane isolation approach in the Methods section. In this case, U138-MG cells seeded at 10^5^ cells in a 60 mm diameter dish were grown in standard conditions until nearly confluent. Following cell lysis, cytosolic and membrane fractions were separated by centrifugation. Plasma membrane fractions were then mechanically fragmented as previously described. Like before, generated M-NVs were analyzed in terms of concentration and diameter sizes by the qNano Gold instrument. 

A fixed amount of M-NVs, i.e., 6 × 10^7^ particles generated by the above-mentioned methods, were subjected to a spontaneous reassembly onto spherical structures using room temperature agitation for an hour and stained with CBG dye for immunofluorescence and cytometric detections.

To compare the kinetics of the internalization of U138-MG-derived M-NVs following osmotic lysis or direct plasma membrane isolation protocols, identical amounts of CBG-stained M-NVs were firstly purified from unstained dye using Exosome Purification Columns and finally administered to U138-MG cells grown in standard conditions. Inverted microscope fluorescent evaluations were performed at different time intervals (i.e., 1–3–24 h). As a result, fluorescent M-NVs were timely and progressively internalized, showing higher cytoplasmic fluorescent spots after 24 h p.t., particularly those generated by direct plasma membrane isolation protocol (Appendix A).

Flow cytometry analysis (Amnis Imagestream) was employed to evaluate the morphological distribution of CBG-stained M-NVs generated through the two reported methods. Comparing scatterplot distributions and their normalized frequencies, M-NVs derived from the osmotic lysis method displayed two sub-populations: the former was exosome-like (R2 gate) and the latter represented a larger heterogeneous in dimension vesicles (R3), respectively. In contrast, M-NVs generated by direct plasma membrane isolation produced mainly a highly enriched homogeneous cluster of exosome-like particles (Figure 1A,B). Particle dimensional analysis and distribution by TRPS analysis by using qNano Gold instrument confirmed an exosomal size range with an average of 120 ± 80 nm for direct plasma-membrane-generated M-NVs compared to higher average dimensional values of 310 ± 160 nm for the osmotic lysis method (Figure 1C). Following these results, further experiments were performed adopting M-NVs generated by the direct plasma membrane isolation approach. 

### 3.2. Quantitative Cellular Internalization Analysis of M-NVs

To evaluate the cellular internalization capability of plasma-membrane-generated M-NVs, a panel of human primary normal fibroblasts and tumor-established cells (i.e., uterine cervix cancer HeLa, high-grade glioma U138-MG and U373-MG) were considered as precursors for the isolation of M-NVs and as further combinatory recipients of the generated biomimetic vesicles. M-NVs (6 × 10^7^ particles) were generated as described above and stained with CBG, purified from unbound dyes, and finally administered to different recipient cells (5 × 10^6^ into 60 mm dishes) equally. After 24 h of incubation, the treated cells were collected and resuspended into 0.22 μm filtered D-PBS (100 μL). By using flow cytometry analysis (Imagestream), cell populations were first visualized as dimensional scatterplot distributions, while M-NV uptake was quantified with the “Cell internalization” wizard (Amnis, Ideas 6.2). Internalization scores related to each R1 scatterplot gate (i.e., 10,000 cells) are reported as R4 plots in Figure 2 and summarized in Table 1. Image example galleries of fluorescent-internalized (R4 gates) and not-cell-internalized M-NVs (R5 gates) are shown in Appendix A. 

According to the internalization data reported above, U138-MG-glioma-generated M-NVs showed generally the highest uptake efficiencies into different recipient cells. Furthermore, U138-MG cells displayed the highest capacity to internalize M-NVs. In addition, strictly homotypic internalization trends were not reported. Among the assayed M-NVs, those derived from human normal fibroblasts reported the lowest internalization values (Table 1). Therefore, the next analyses were focused on U138-MG-generated M-NVs.

### 3.3. Ultrastructural and Molecular Characterization of U138-MG-Derived M-NVs

Subsequently, to characterize the morphological and molecular features of M-NVs, U138-MG-derived nanovesicles were analyzed by TEM. The likely round-shaped particles of nanometric sizes characterized by double membrane layers were highlighted (Figure 3).

Next, immunoblotting analysis was performed on protein extracts of U138-MG-derived M-NVs to evaluate the presence of extracellular vesicle markers. As expected, CD63, CD81, and Annexin V showed a marked expression. Additionally, CD47 expression was highlighted, while the cytosolic protein Hsp70, expected as the negative control, was not scored into M-NV-derived proteins (Figure 4).

### 3.4. Cytotoxicity Evaluation of U138-MG-Derived Empty M-NVs and Transfection Assays with DNA/Proteins/Drug Cargoes

An MTT cellular vitality assay was performed to verify if plasma-membrane-derived nanovesicles without any specific molecular cargo might induce cytotoxicity per se. Thus, cargo-less M-NVs, derived from the plasma membrane of U138-MG cells, were administered to U138-MG cells (6 × 10^7^ particles/10^5^ cells); then, an MTT vitality assay was performed in untreated (NT) vs. treated cells at 24 and 48 h p.t. showing no significant differences in viability.

To investigate the ability of M-NVs to host nucleic acids and deliver to recipient U138-MG cells, U138-MG-derived M-NVs (6 × 10^7^ particles) were loaded with a pEGFP-expressing plasmid. Identical amounts of the vector (i.e., 1 μg) were transfected to the same cells by electroporation or lipofectamine-based protocols [28] and compared. As reported in Appendix A, no evident differences in the transfection approaches were recorded. 

As widely reported, the immunodetection of cytochrome c requires the selective permeabilization of plasma membranes [35]. To evaluate the capability of M-NVs to shuttle impermeant molecules within cells, an anti-cytochrome c DyLight450 conjugated antibody was directly administered to growing U138-MG cells and visualized by inverted fluorescent microscope at 48 h p.t. In addition, cells were also stained with a cell-permeant dye for mitochondria visualization (i.e., MitoTracker Deep Red, 0.1 μM for 15 min before visualization). As reported in Figure 5 (left panel), the anti-cytochrome c DyLight450 antibody failed to penetrate plasma membranes, thus resulting in green signals mostly localized at the outer plasma membranes, in contrast to the mitochondrial intracellular staining. On the other panel, the same cells were incubated for 48 h with U138-MG-originated M-NVs (6 × 10^7^ particles) assembled with the same amount of cytochrome c DyLight450 antibody. As a result, yellow intracellular fluorescent spots were highlighted, likely produced by the antibody released in the cytoplasm and to the further co-localization with the red mitochondrial signals, along with a decrease in the pericellular staining of cytochrome c DyLight450 antibody signals (Figure 5, right panel).

Finally, M-NVs derived from U138-MG plasma membranes were produced with Rose Bengal (RB) as cargo. RB is an auto-fluorescent photosensitizer with a relatively low efficiency to cross plasma membranes [36]. Hence, U138-MG cells (10^5^ into 30 mm dishes) were treated with RB (12.5 μM) and internalization was monitored at 24 and 48 h p.t. In parallel, U138-MG cells were incubated with M-NV (6 × 10^7^ particles) encapsulated with the same RB concentration. Following time kinetics, RB was minimally internalized as a direct administration scheme, while massive uptake was highlighted in the case of RB encapsulated by M-NVs (Figure 6).

### 3.5. Cytotoxicity Evaluation of U138-MG-Derived RB M-NVs 

According to the above results indicating an effective RB uptake mediated by M-NVs, intracellular RB content was subjected to light irradiation stimulation to potentially increase RB-induced cytotoxicity as reported previously [29]. Thus, U138-MG cells were tested by direct RB administration in cell culture media (12.5 μM), followed by light irradiation (562/576 nm, 5 min), and were finally evaluated at 72 h p.t. This condition was compared with the same cell concentration (10^5^ into 30 mm dishes), incubated with M-NVs (6 × 10^7^ particles) containing RB cargo at the same molar concentration, and subjected to an identical light irradiation protocol. As documented in Figure 7, optical microscopy evaluations highlighted a prominent toxicity for an M-NVs (RB) + light irradiation combined scheme, while no evident viability effects were scored for direct RB administration and light irradiation treatment alone. Furthermore, the RB fluorescence microscope analysis of the former treatment scheme showed a marked intracellular accumulation of RB, particularly localized within nuclei.

Membrane fractions isolated from U138-MG cells were lyophilized and de novo tested after long-term conservation. Thus, lyophilized membrane fractions were reconstituted in PBS at the same starting concentration and re-assembled with RB (12.5 μM) for 1 h at room temperature under agitation. The ultrastructural evaluation of the obtained M-NVs with RB cargo was performed by TEM and size-analyzed by TRPS technology, showing a round shape and exosomal diameter ranges similar to empty M-NVs (Appendix A). The internalization capability of the reconstructed M-NVs (RB) was assayed at 24 h p.t. in U138-MG cells (6 × 10^7^ particles/10^5^ cells into 30 mm diameter dishes) and documented (Appendix A). To evaluate cytotoxicity effects in normal (rat primary astrocytes) vs. tumor cells (U138-MG), empty M-NVs and RB-reconstructed M-NVs, i.e., M-NVs (RB), were analyzed by MMT viability assay at different time intervals (24, 48, and 72 h p.t., using above-mentioned conditions). As reported in the graph (Appendix A), a viability reduction of about 50% was scored after 72 h p.t. only in glioma cells. Differently, normal rat astrocytes, treated with empty M-NVs or with RB-loaded counterparts, did not show cytotoxicity effects in the investigated time interval.

Finally, M-NVs were administered to U138-MG cells (6 × 10^7^ particles/10^5^ cells in 30 mm diameter dishes) grown and compared to RB (12.5 μM)-treated cells. At 48 h p.t., a higher RB florescence internalization by reconstituted M-NVs compared to directly administrated RB was reported, alongside an increase in cytotoxicity even in the absence of light irradiation (Figure 8A). Then, to confirm and quantify these data, a long-term clonogenic assay was performed. Following 48 h of incubation, U138-MG cells subjected to RB (12.5 μM), RB-encapsulated lyophilized M-NVs (12.5 μM into 6 × 10^7^ particles), and untreated cells were trypsinized and re-seeded at 1000 cells into 60 mm dishes and grown for one week; then, cells were fixed with ethanol and stained with Crystal Violet (0.5%). Cell clones were visualized using a Nikon Eclipse TS100 inverted microscope (4× magnification). The results showed differences in the amount of colonies; in particular, U138-MG cells that were treated with reconstituted M-NVs (RB) displayed a significant reduction compared to only RB-treated and untreated cells (Figure 8B).

## 4. Discussion

When comparing different brain cancers, GBM is undoubtedly considered to be the most aggressive form, characterized by poor prognosis and high mortality [37]. These clinical features can be explained by an intrinsic high heterogeneity and complexity of the tumor itself, as well as the difficulty in reaching an effective concentration of the drugs within the brain lesion because of the tight junction complexes that form BBBs. Consequently, a plethora of pharmacological nanocarriers have been assayed in recent years for their enhanced specificity and efficiency in delivering anticancer agents [38]. Among these, biomimetic or cell-membrane-camouflaged nanoparticles, made by biological membranes and hosting anticancer agents, have been demonstrated to sustain long-term circulation, be able to escape immune responses, and target cancer cells efficiently [39]. Membrane templates, useful for creating biomimetic or “ghost cells” or “reduced protocells” vectors, can be derived from cell extrusion, mechanical sonication, or through direct microfluidics devices [40]. By adopting these generating approaches, molecular heterogeneous cargoes such as siRNA, mRNA, and proteins can be mixed with plasma membrane components and intraluminarly vesiculated [40]. To simplify these technical approaches, the principle of the self-assembly of discrete isolated fragments of plasma membranes into spherical structures with the contemporary encapsulation of the surrounding experimentally provided molecules (i.e., nucleic acids, proteins, drugs) into the lumen cavity was adopted [41].

Among the protocols employed to produce membrane biomimetic nanovesicles based on osmotic lysis or direct plasma membrane isolation, the latter demonstrated better performances in generating vesicles of dimensions in the exosomal range. This morphological feature, highlighted by TEM, flow cytometry, and TRPS analysis, might be relevant in permitting a more favorable dimensional range of the generated nanovesicles for crossing biological barriers such as the BBB [42].

Also, through the generation of different M-NVs from normal and cancer plasma membranes, coupled with combinatorial cellular internalization fluorimetric assays, U138-MG-generated M-NVs and their parental glioma cells were further investigated in terms of their higher capability to host and specifically deliver nucleic acids, proteins, and drugs. Notably, the molecular analysis of the U138-MG-generated M-NVs highlighted the presence of CD47; this glycoprotein is crucial in preventing the direct clearance of the same vesicles by macrophages, allowing an increase in bioavailability, even in the context of GBMs [43].

The investigation of homotypic and heterotypic interactions by studying the cellular uptake of homologous M-NVs (i.e., donor plasma membrane components and recipient cells of the same type) vs. heterologous ones (donor plasma membrane components and recipient cells of different types) did not reveal particular differences in uptake yields within combinations deriving from cancer cells; in contrast, M-NVs derived from normal primary fibroblasts showed a reduced internalization capability in all recipient cells. Scored differences in M-NV uptake, particularly the results showing a greater capability of glioma-derived cells (i.e., U138-MG and U373-MG) in internalizing homologous and heterologous nanovesicles, might be related to the presence of specific surface molecules, both in the case of plasma membranes and generated vesicles. These moieties, particularly CD169 (sialoadhesin) and heparin sulfate proteoglycans, were demonstrated as playing a role in capturing exosomes in glioblastoma cells [44]. Furthermore, the greater internalization scored by cancer cells might be related to the low pH of the tumor environment and hypoxic conditions that promote an increased delivery and uptake of extracellular vesicles into the tumor cells [45,46]. 

Among the assayed incorporated molecules into the generated vesicles, RB was widely reported to display a relatively low cellular uptake when directly administered [47]. In contrast, M-NVs with RB cargo showed a marked internalization of the compound, leading to a significant decrease in viability following photo-stimulation, similar to those obtained in glioma cells produced by higher RB concentrations [29]. 

Previous studies have highlighted that RB induces apoptosis in tumor cells even in the absence of photo-stimulation. The mechanisms through which RB alone can induce selective cytotoxicity are not yet fully understood. However, the effects may be due to its ability to affect various organelles such as the mitochondria by producing ROS or nucleus by the induction of irreparable DNA ruptures when highly internalized within a cell [48]. This evidence might provide a direct interpretation of the obtained results with M-NVs loaded with RB that induced cytotoxicity even without light stimulation, showing marked intracellular and nuclear RB staining.

In addition, the functionality of U138-MG-generated M-NVs in terms of molecular cargo loadings, efficient cellular internalization, and the induction of the cytotoxic effect was demonstrated by using vesicles subjected to a lyophilization protocol. It was reported that drug-loaded nanoparticles were sensitive to physical and chemical instability, forming larger aggregates that, altogether, produced a decrease in functionality. There are various ways to stabilize such nanoparticle-based formulations, including the addition of ionic materials to provide electrostatic repulsion or polymer materials forming a steric barrier between the particles. However, for long-term stability, water often needs to be removed to obtain a dry product [49]. 

In conclusion, this preliminary contribution presents limitations mostly related to the functional evaluation of the therapeutic efficacy of M-NVs in further experimental models, particularly those involving animal models of induced glioblastoma. Primarily, in the definition of M-NV administration as a systemic scheme or mediated by intratumor nanodevices with the direct possibility to perform a light irradiation stimulation or utilizing intranasal protocols, it has been reported that therapeutic exosomes can reach specific brain regions [50]. Also, independently of administration routes, the reported expression of CD47 by M-NVs needs to be challenged in terms of macrophage-reduced uptake and clearance. In relation to this consideration, deeper investigations on the stealth properties of the generated M-NVs will be required. Importantly, it was reported that the stealth effect, defined as the ability of nanomaterials to be invisible to the immune system, leading to reduced clearance and enhanced retention in the bloodstream, is a crucial feature for drug delivery nanomaterials [51]. These evaluations in terms of surface structure, geometry, minimal or dipole charges, and the presence of the hydrophobic domains of the generated M-NVs will be very helpful in defining and eventually improving pharmacokinetics of the bio-inspired nanovesicles, as generated from plasma membrane components. Another concern to be considered is that the vascular permeability of the brain is well known to be extremely low due to the blood–brain barrier (BBB), only allowing the transport of some small molecules (generally inferior to 500 Da) [51]. In particular, it is reported that most small-molecule drugs can passively permeate by transcytosis through a sieve-like vascular of the BBB [52], strengthening the possibility that the nanoparticles generated in this study, with exosomal dimension ranges, can favorably transit through the BBB. 

The clinical adoption of cells’ derivatives as biomimetic drug-loaded nanovesicles for brain tumor therapeutics is still in a primordial experimental phase. Crucial drawbacks need to be clarified before this type of strategy can be translated into clinical trials. Firstly, by means of in vivo studies, the deciphering of the organ/tissue/cell tropism of the vectors, as well as the interactions of the nanocarriers with non-neoplastic cells, is crucial for reducing off-target effects and adverse immune responses. Accordingly, the evaluation of the entire manufacturing chain in terms of purification strategies, storage, stability, drug loading efficiency, and dosage are fundamental to guarantee reproducible therapeutic outcomes. Additional preclinical translational concerns related to the scalability and reproducibility of biomimetic production must be rigorously addressed. For the first aspect, the possibility of expanding cellular material (coming from primary or stabilized tumor lines) in appropriate bioreactors to isolate membrane templates for the generation of biomimetic vesicles could guarantee adequate scalability in larger production and subsequent use. For the latter, a further refinement of size to maintain their nanosize advantage, molecular composition, cargo content, and long-term in vivo stability is required, and the drug release properties of biomimetic nanovesicles will require standardization in order to guarantee consistent and reproducible outcomes for brain tumor therapeutic interventions.

## Figures and Tables

**Figure 1 nanomaterials-14-01779-f001:**
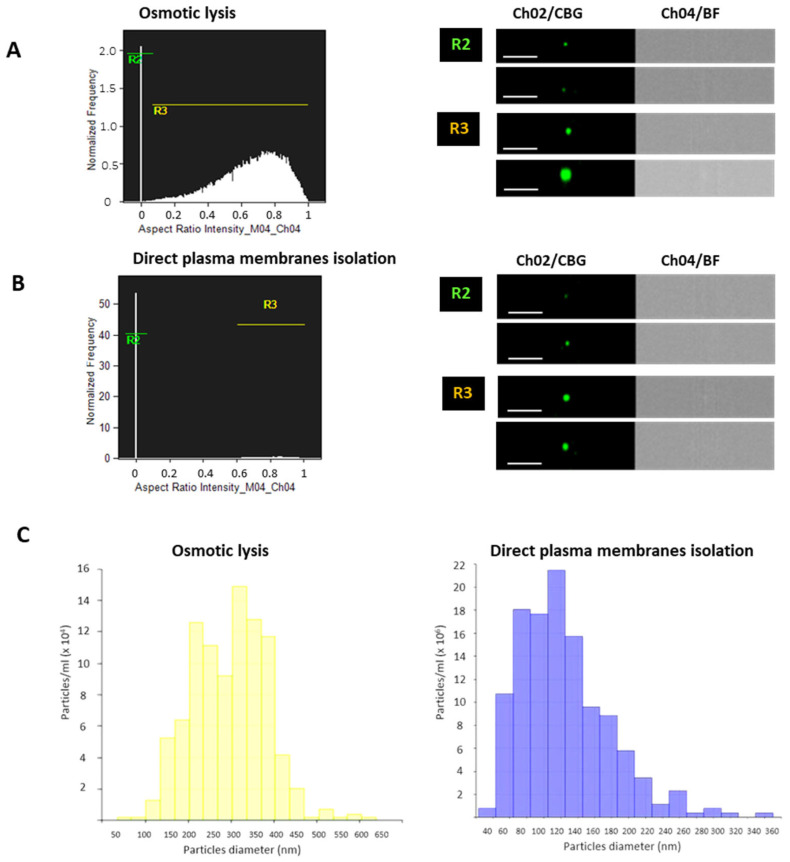
Flow cytometry and TRPS analysis of M-NVs. M-NVs were derived from osmotic lysis (**A**) or the direct plasma membrane isolation approach (**B**) from U138-MG cells, stained with CBG, and analyzed by Amnis Imagestream flow cytometry (60× High Gain Mode). M-NV (*n* = 10,000) size distribution resulted in two gates (i.e., R2 and R3) and likely represented exosome ranges (40–140 nm, R2) and larger diameter particles (150–600 nm, R3), as revealed by TRPS analysis performed with the qNano Gold instrument (*n* = 2000) (**C**). CBG fluorescent (Channel 02, Ch02) and Bright field (BF, Ch04) particles are visualized in the image galleries. Scale bars = 1 µm.

**Figure 2 nanomaterials-14-01779-f002:**
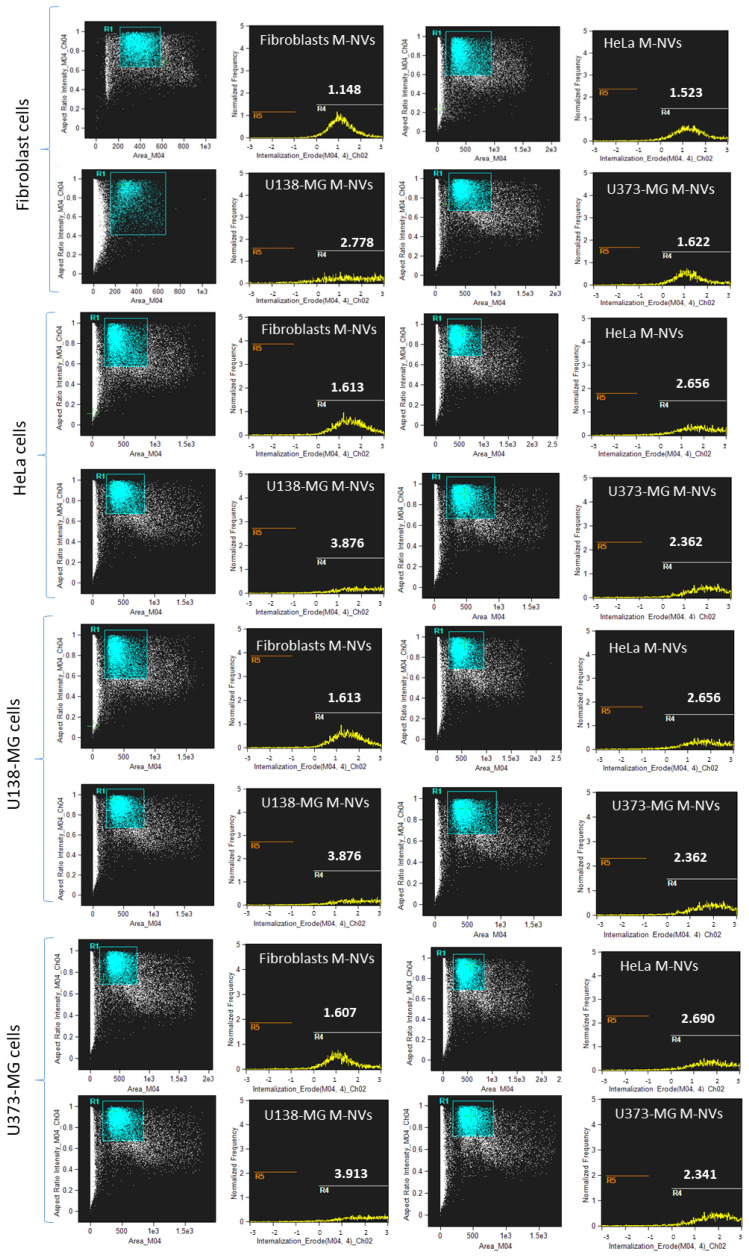
Flow cytometry M-NV internalization assays. M-NVs were isolated from different cells by the direct plasma membrane isolation approach, quantified using qNano Gold instrument (6 × 10^7^ particles), stained with CBG dye, administered to equal amounts of human primary fibroblasts, HeLa, or glioma cells (i.e., U138-MG and U373-MG), and finally analyzed using flow cytometry (Amnis Imagestream). The “Internalization” wizard was adopted, where the scored R4 gates represented internalization values of fluorescent M-NVs, while R5 indicated cells without M-NV uptake. R1 indicated a scatterplot of focused cells (*n* = 10,000 events). Representative image galleries are reported in Appendix A. A summary of R4 internalization scores is schematized in Table 1.

**Figure 3 nanomaterials-14-01779-f003:**
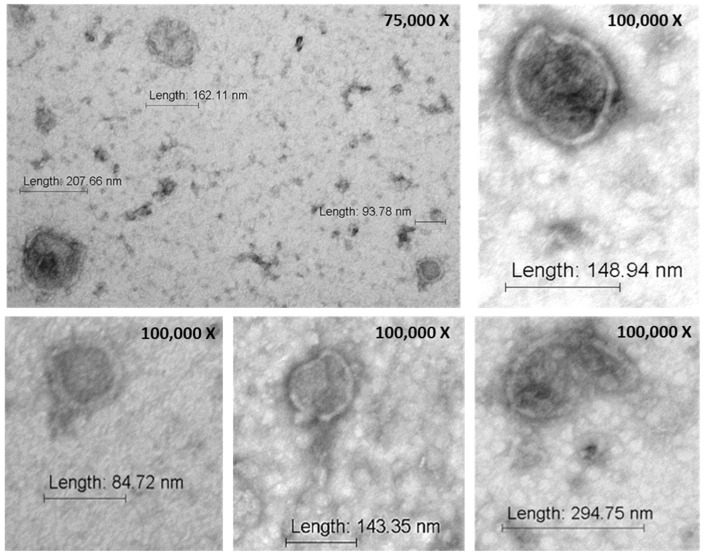
Ultrastructural analysis of M-NVs. M-NVs were derived from the direct plasma membrane isolation approach of U138-MG cells and analyzed in terms of morphology and diameter by TEM.

**Figure 4 nanomaterials-14-01779-f004:**
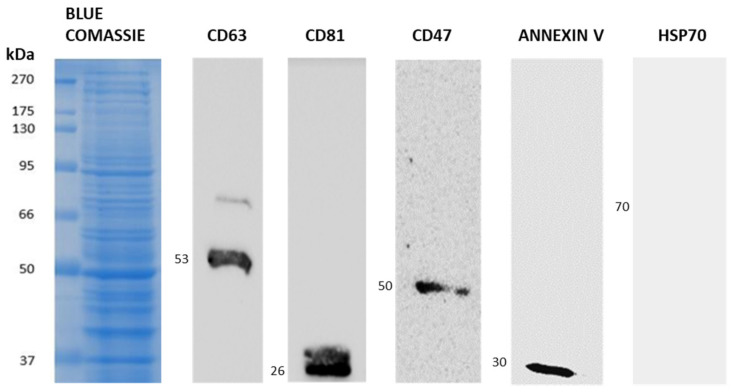
Protein immunoblotting analysis of M-NVs. M-NVs were derived by the direct plasma membrane isolation approach from U138-MG cells, quantified in their total protein content, and analyzed by immunoblotting. Molecular weights are reported as kDa.

**Figure 5 nanomaterials-14-01779-f005:**
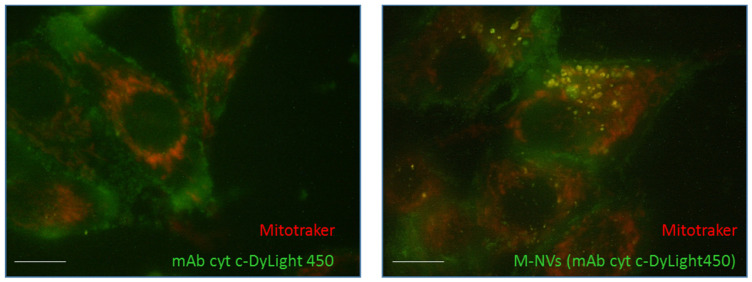
Immunofluorescence internalization of M-NVs with an impermeant protein cargo. U138-MG cells were incubated with fluorescent-conjugated DyLight450 anti-cyt c mAb in the absence of cellular permeabilization and visualized by inverted fluorescence microscope (Nikon Eclipse TS100 100× oil immersion objective using B2A filter) at 48 h p.t. (**left panel**). In parallel, M-NVs derived by direct plasma membrane isolation from U138-MG cells, quantified by qNano Gold instrument (6 × 10^7^ particles), incubated with an identical amount of fluorescent-conjugated antibody, were administered to U138-MG cells (**right panel**). Cells were also stained with MitoTracker Deep Red and visualized. Scale bars = 10 µm.

**Figure 6 nanomaterials-14-01779-f006:**
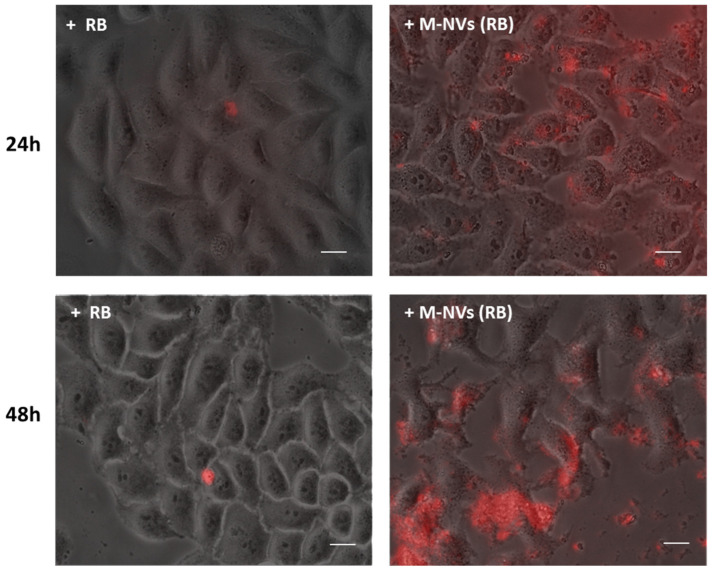
Immunofluorescence evaluation of RB or RB encapsulated by M-NV internalization. U138-MG cells were treated with RB (12.5 µM) and visualized by inverted fluorescence microscopy at 24 and 48 h p.t. (**left panels**, Nikon Eclipse TS100, 40×, G2A filter). An identical amount of RB was in parallel incorporated into M-NVs (6 × 10^7^ particles), derived by the direct plasma membrane isolation approach from U138-MG cells, and similarly evaluated (**right panels**). Scale bars = 10 µm.

**Figure 7 nanomaterials-14-01779-f007:**
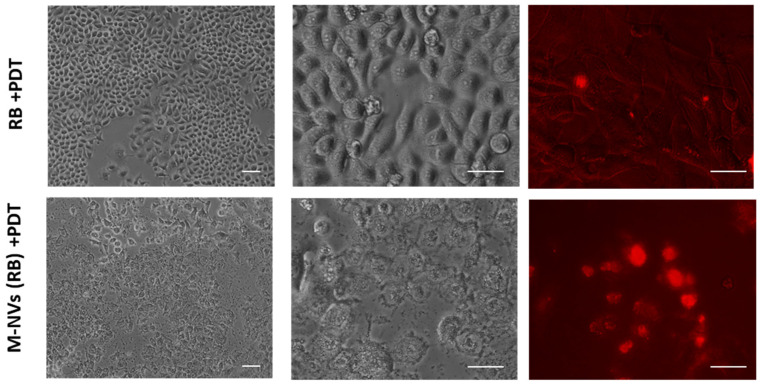
Evaluation of RB or RB-mediated M-NV cytotoxicity in combination with light irradiation stimulation. U138-MG cells were treated with RB (12.5 µM) and visualized by optical (10 and 40×) inverted fluorescence (Nikon Eclipse TS100, 40×, G2A filter) microscopy at 72 h p.t. (**upper panels**). An identical amount of RB was simultaneously incorporated into M-NVs (6 × 10^7^ particles), derived by the direct plasma membrane isolation approach from U138-MG cells, and similarly evaluated (**lower panels**).

**Figure 8 nanomaterials-14-01779-f008:**
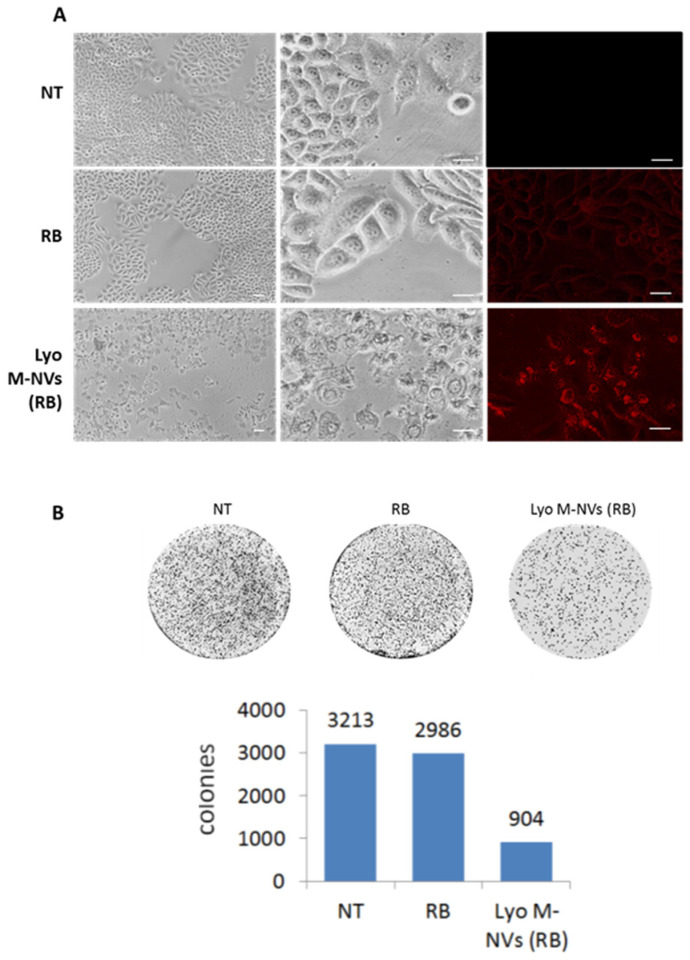
Long-term cytotoxic effect of lyophilized M-NVs loaded with RB. (**A**) U138-MG cells were treated with RB (12.5 µM) and visualized by optical (10 and 40×) inverted fluorescence (Nikon Eclipse TS100, 40×, G2A filter) microscopy at 48 h p.t. (**upper panels**). An identical amount of RB was in parallel incorporated into M-NVs (6 × 10^7^ particles), derived by the direct plasma membrane isolation approach from U138-MG cells. (**B**) A clonogenic assay on 1000 cells was performed for one week on U138-MG untreated cells and cells treated with RB (12.5 µM) or with the same RB amount incorporated into M-NVs (6 × 10^7^ particles, derived by the direct plasma membrane isolation approach from U138-MG cells and lyophilized as described in Section 2). Colonies were stained with Crystal Violet and counted with ImageJ. Scale bars = 20 µm.

**Table 1 nanomaterials-14-01779-t001:** Flow cytometry cell internalization indexes of differently generated M-NVs.

Recipient Cells	M-NVs	M-NVs	M-NVs	M-NVs
	Normal fibroblasts	HeLa	U138-MG	U373-MG
Normal fibroblasts	1.148	1.523	2.778	1.622
HeLa	1.163	2.656	3.876	2.362
U138-MG	1.162	3.168	3.665	3.127
U373-MG	1.607	2.690	3.913	2.341

Values are derived from internalization measurements of R4 gates as illustrated in Figure 2.

## Data Availability

Data are contained within the article or Appendix A.

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
