# Peer review of "Development and Biological Characterization of Cancer Biomimetic Membrane Nanovesicles for Enhancing Therapy Efficacy in Human Glioblastoma Cells"

_nanomaterials, 2024, doi:10.3390/nano14221779_

Round 1

Reviewer 1 Report

Comments and Suggestions for Authors

The authors showed cancer biomimetic membranes nanovesicles as efficient drug delivery strategy for human glioblastoma cells. Overall, it is suitable for publication in this journal. But before publication, please address the following concerns.

1. Comprehensive characterization of RB-loaded M-NVs, including size, TEM, loading efficiency and content, stability, and drug release profile.

2. More quantitative results should be shown in the main text, such as fluorescence intensity from microscopy and IC50.

3. There are two major advantages of biomimetic or cell membrane camouflaged nanoparticles, providing stealth properties and at the same time keeping cancer cell-targeting ability. This point should be emphasized more clearly in the paper (https://doi.org/10.1016/j.addr.2023.114895).

4. The wording should be more accurate. For example, in most cases, 'PDT' in this paper should be replaced by 'light irradiation'. PDT already means photodynamic therapy. Also please check the abbreviation to keep accurate.

5. Regarding stealth properties, could you provide some evidence such as in vitro macrophage uptake? 

Author Response

AUTHORS’ REPLY REFEREE 1

The authors showed cancer biomimetic membranes nanovesicles as efficient drug delivery strategy for human glioblastoma cells. Overall, it is suitable for publication in this journal. But before publication, please address the following concerns.

  1. Comprehensive characterization of RB-loaded M-NVs, including size, TEM, loading efficiency and content, stability, and drug release profile.

R1: Part of these data have been included in a novel Supplementary Figure S5, specifically nanoparticles (M-NVs loaded with RB) size by TRSP by using qNANO Gold instrument with the identical setting of those adopted for Figure 1. As a comment, size of the novel assayed M-NVs (RB) were roughly similar to empty M-NVs generated from U138 cells with the direct plasma membrane isolation and reconstruction described protocol. The same RB loaded nanoparticles, i.e. M-NVs (RB) were also exanimated by TEM, as illustrated in Figure S5 panel A highlighting similar round-like defined morphology and sizes in agreement with those revealed by TRSP analysis (panel B). Honestly, important but also complex request additional pharmacokinetics studies, as loading efficiency, stability and drug release profiles were not performed in this contribution, but these will be for sure included in a consequent continuation of the study.

Also, hoping the Referee will understand, the Journal imposed a very limited and non-extendable time frame (not over 2 weeks !) for providing responses to the Referees, realistically not feasible to perform these studies.

  1. More quantitative results should be shown in the main text, such as fluorescence intensity from microscopy and IC50.

R2: In agreement with the Referee, we have provided novel immunofluorescence microscopy evaluation of RB loaded M-NVs in U138-MG cells in a short post-treatment interval (i.e. 24 hours, Figure S5 C) to better evaluate internalization and cytoplasmic distribution of the fluorescent nanoparticles, deliberately avoiding long time intervals analysis that will induce cytotoxic effects as reported in the manuscript (Figure 7, lower panels). IC50 data were provided considering MTT viability assays in normal rat astrocytes vs U138-MG human glioma cells, assaying empty M-NVs (using identical conditions reported in the manuscript) and RB-loaded M-NVs (12.5 mM), and evaluating viability trends at 24-48 and 72 hours p.t. As reported in the graph (Figure S5 panel D) a viability reduction of about 50% was scored after 72 hours p.t. only in glioma cells. Differently, normal rat astrocytes, treated with empty M-NVs or with RB-loaded counterparts, did not show cytotoxicity effects along the investigated time interval

  1. There are two major advantages of biomimetic or cell membrane camouflaged nanoparticles, providing stealth properties and at the same time keeping cancer cell-targeting ability. This point should be emphasized more clearly in the paper (https://doi.org/10.1016/j.addr.2023.114895).

R3: Authors thanks Referee for this important suggestion that has been introduced within the revised manuscript in the last part of the discussion including related reference

  1. The wording should be more accurate. For example, in most cases, 'PDT' in this paper should be replaced by 'light irradiation'. PDT already means photodynamic therapy. Also please check the abbreviation to keep accurate.

R4: This important and proper use of the suggested terminology has been corrected in the manuscript

  1. Regarding stealth properties, could you provide some evidence such as in vitro macrophage uptake? 

R5: The question of the Referee is of course very pertinent and precise and this will be one of our next starting point for a consequent continuation and experimental investigation. Again, even if we are stimulated in carrying out and evaluating the behavior in an in vitro system of our nanovesicles in the presence of macrophages, the immediacy required in completing the evaluation round requires us to consider this experimental aspect in a subsequent contribution.

Reviewer 2 Report

Comments and Suggestions for Authors

The present article entitled "Development and Characterization of Cancer Biomimetic Membranes Nanovesicles as Efficient Drug Delivery Strategy for Human Glioblastoma Cells" from Massarotti et al provides a clear rationale for developing biomimetic nanovesicles (M-NVs) as a novel drug delivery system, especially for human glioblastoma treatment. It highlights the limitations of current therapeutic methods, such as the blood-brain barrier (BBB) challenge and low efficacy of direct drug delivery. The discussion of extracellular vesicles (EVs), their classification, and their use in drug delivery is relevant and backed by appropriate references.

Suggestions/Improvements/questions:

  • The introduction could benefit from a more detailed comparison of M-NVs to other nanocarriers (e.g., polymeric nanoparticles, lipid-based systems) to highlight their advantages and limitations better.
  • The clinical relevance of M-NVs should be more explicitly connected to the broader therapeutic landscape for glioblastoma. How do the authors envision this technology being integrated into current clinical practice?
  • Is there any preliminary in vivo evidence or data that supports the claims made for M-NVs? Including this would strengthen the state of the art.
  • Can the authors expand on the specific challenges of delivering drugs through the BBB with respect to this study?

The methods are clearly explained, with two techniques used for M-NV generation: osmotic lysis and direct plasma membrane isolation. Detailed protocols for the generation, staining, and molecular cargo loading of M-NVs are provided, which allows for reproducibility.

Suggestions/Improvements/questions:

  • While the methods are comprehensive, the authors should mention why they selected the specific glioma cell lines (U138-MG, U373-MG) over other potential models.
  • The parameters used for electroporation and lipofection in transfection assays should be justified. Why were these specific settings chosen?
  • Did the authors consider other methods of producing M-NVs, such as mechanical extrusion or microfluidics, and how do these compare to the methods used in this study? How was the efficiency of each M-NV generation method quantitatively assessed? While TRPS and flow cytometry were used, a more explicit comparison of the yields between the two methods would be helpful.

Moreover, the results are systematically presented, comparing M-NVs produced via the two different methods. The study shows that M-NVs derived from direct plasma membrane isolation are more homogeneous and exhibit a smaller size range, which is a favorable trait for drug delivery applications.

Suggestions/Improvements/questions:

  • A deeper discussion on the differences in internalization between homotypic and heterotypic interactions (e.g., why U138-MG cells have a higher uptake) would add value to the findings.
  • Can the authors provide more quantitative data on the reproducibility of the M-NV generation process? How variable were the results across multiple batches?

Furthermore, the discussion is appropriate and contextualizes the findings within the field of glioblastoma research and nanomedicine. It appropriately highlights the potential of M-NVs to enhance the delivery of drugs like Rose Bengal, especially in overcoming limitations related to low uptake and off-target effects.

Suggestions/Improvements/questions:

  • The authors should address the potential immune response to M-NVs in vivo. While CD47 expression provides some immune evasion, there may still be challenges with long-term circulation and clearance.
  • Consider adding a section that discusses potential off-target effects of M-NVs in non-tumor tissues. Are there concerns about unwanted cytotoxicity in healthy brain cells?
  • Given the focus on glioblastoma, have the authors considered testing M-NVs in more aggressive or resistant glioma subtypes?
  • How scalable is the production method for M-NVs, and what are the next steps to move this from a lab-based study to clinical trials?

Finally, the conclusions summarizing the findings that M-NVs show promise as drug delivery systems for glioblastoma treatment. However, the long-term therapeutic efficacy, especially in in vivo models, remains to be demonstrated.

Suggestions/Improvements/questions:

  • The authors should briefly outline the limitations of their current study in the conclusion and propose future directions.
  • Have the authors considered investigating the pharmacokinetics of M-NVs in vivo?
  • Are there any plans to explore potential combination therapies using M-NVs (e.g., with standard chemotherapy agents)?

Overall recommendation: Minor revisions. The study is innovative and offers a significant contribution to the field, but addressing the above points will improve its relevance.

Author Response

AUTHORS’ REPLY REFEREE 2

REPORT 2

Comments and Suggestions for Authors

The present article entitled "Development and Characterization of Cancer Biomimetic Membranes Nanovesicles as Efficient Drug Delivery Strategy for Human Glioblastoma Cells" from Massarotti et al provides a clear rationale for developing biomimetic nanovesicles (M-NVs) as a novel drug delivery system, especially for human glioblastoma treatment. It highlights the limitations of current therapeutic methods, such as the blood-brain barrier (BBB) challenge and low efficacy of direct drug delivery. The discussion of extracellular vesicles (EVs), their classification, and their use in drug delivery is relevant and backed by appropriate references.

Suggestions/Improvements/questions:

  • The introduction could benefit from a more detailed comparison of M-NVs to other nanocarriers (e.g., polymeric nanoparticles, lipid-based systems) to highlight their advantages and limitations better.
  • R: This important improvement has been made in the introduction section, along with relative references

  • The clinical relevance of M-NVs should be more explicitly connected to the broader therapeutic landscape for glioblastoma. How do the authors envision this technology being integrated into current clinical practice?
  • R: A preliminary perspective, that was included in the current limitations and future challenges section, at the end of the revised discussion version, will try to provide some hypothetical scenario into clinical investigations.
  •  
  • Is there any preliminary in vivo evidence or data that supports the claims made for M-NVs? Including this would strengthen the state of the art.

R: Even if, as explained in some answers, we have no direct data on in vivo experiments, that, as already stated, will be considered in our next plans, as highlighted in the limitation of the study section within the discussion section

  • Can the authors expand on the specific challenges of delivering drugs through the BBB with respect to this study?
  • R: Again, this is a relevant perspective, strictly related to one of the main obstacles in developing efficient therapeutic strategies in glioma strategies. We haven’t performed yet related assays, for examples those can mimic in vitro cellular tight junctions barriers and, neither of course in vivo BBB assays. Consequently, in agreement with topic importance underlined by the Referee, we have provided integratory parts from scientific literature on this topic in the discussion (limitation and future perspective).

The methods are clearly explained, with two techniques used for M-NV generation: osmotic lysis and direct plasma membrane isolation. Detailed protocols for the generation, staining, and molecular cargo loading of M-NVs are provided, which allows for reproducibility.

Suggestions/Improvements/questions:

  • While the methods are comprehensive, the authors should mention why they selected the specific glioma cell lines (U138-MG, U373-MG) over other potential models.
  • R: The following technical justification on the selection of U138-MG, U373-MG over other potential cell lines can be provided: the above mentioned glioma cells were selected according to the different morphological features, the former displaying a polygonal-like regular shape, the latter with a typical astrocyte-like morphology. We believed to consider these nearly opposite cells architectures considering that the proposed methods to generate nanovesicles is based on the isolation of outer membranes. In analogy, other glioma established cell lines might be representative of the two shapes, as T98G for the former and, for example, U87-MG for the latter.

  • The parameters used for electroporation and lipofection in transfection assays should be justified. Why were these specific settings chosen?

R: We completely agree with the criticism related. We can immediately answer that in general both transfection methods, Lipofectamine and electroporation, were adopted by applying mild parametric conditions, to avoid induction of off target cytotoxicity. Specifically, Lipofectamine amount was set in a minimal amount of the manual suggested values (i.e. 5 ml into a 30 mm dish) to induce transfection of plasmids as recommended by Invitrogen but avoiding excessive lipids vesicles cytoplasmic accumulation. Similarly, electroporations conditions (i.e., ΔV = 600 V, pulse width = 10 ms, pulse number = 20) was set to not induce excess of cellular damage. These mild conditions of the two methods, even they might not produce the maximal transfection efficiencies, can maintain a similar cellular architecture comparable with the cell sample subjected to a direct administration of M-NVs, loaded with the same quantity of expression plasmid. The missing methodological information related to the amount of Lipofectamine was introduced into 2.3 methods chapter.

  •  
  • Did the authors consider other methods of producing M-NVs, such as mechanical extrusion or microfluidics, and how do these compare to the methods used in this study? How was the efficiency of each M-NV generation method quantitatively assessed? While TRPS and flow cytometry were used, a more explicit comparison of the yields between the two methods would be helpful.

R: We haven’t the possibility to directly test dedicated devices as mechanical extrusion or microfluidics apparatus. We believe they might be suitable in generated M-NVs as widely reported in literature. Due to the lack of these devices, we have focused on a more practical and perhaps easy approaches in generated M-MNs as direct plasma membranes isolation approaches. As stressed within the manuscript we have based our evaluation process of efficiency in generating M-NVs following cellular internalization and particularly on M-NVs geometry in terms of dimension. Since, thinking on futuristic clinical practices, relatively short dimensional rates (i.e. those typical of exosomes, 30-140 nm) have been demonstrated as a favorable feature in permitting BBB crossing, M-NVs generated by direct plasma membranes isolation, appeared to be the nanovesicles of greatest interest for future applications, first in vivo and possibly at a clinical therapeutic level. Flow cytometry (in particular Amnis Imagestream technology) and TRPS can be compared in particular in the dimensional evaluation, in terms of distribution of the nanovesicles the first and in a more analytical way the second, highlighting for the latter the sizing of the particles. According to our results, the yields of the two technology in terms of dimensional analysis of the generated particles with the two adopted protocols can be considered concordant. This concordance in the results, might also explain differences in cellular uptake of the membranes isolating approaches (Figure S2).

Moreover, the results are systematically presented, comparing M-NVs produced via the two different methods. The study shows that M-NVs derived from direct plasma membrane isolation are more homogeneous and exhibit a smaller size range, which is a favorable trait for drug delivery applications.

Suggestions/Improvements/questions:

  • A deeper discussion on the differences in internalization between homotypic and heterotypic interactions (e.g., why U138-MG cells have a higher uptake) would add value to the findings.

R: This topic was deeper exanimated into Discussion section

  • Can the authors provide more quantitative data on the reproducibility of the M-NV generation process? How variable were the results across multiple batches?
  • R: We have periodically cheeked the capability of frozen aliquots of generated M-NVs to be next employed in further administration to cells. For example, as requested by the other Referee, we have performed additional experiments with RB-M-NVs for the described internalization and ultrustructural evaluations reported in panels A and B of Supplementary Figure 5. Even if the time interval from their original generation was several months ago, we think the biological feature of our M-NVs are maintained even from different prepared batches generated from the same cell line membranes (i.e. U138-MG).

Furthermore, the discussion is appropriate and contextualizes the findings within the field of glioblastoma research and nanomedicine. It appropriately highlights the potential of M-NVs to enhance the delivery of drugs like Rose Bengal, especially in overcoming limitations related to low uptake and off-target effects.

Suggestions/Improvements/questions:

  • The authors should address the potential immune response to M-NVs in vivo. While CD47 expression provides some immune evasion, there may still be challenges with long-term circulation and clearance.
  • R: The question of the Referee is of course very pertinent and precise and this will be one of our next starting point for a consequent continuation and experimental investigation. Again, even if we are stimulated in carrying out and evaluating the behavior in an in vitro system of our nanovesicles in the presence of macrophages, the immediacy required in completing the evaluation round requires us to consider this experimental aspect in a subsequent contribution.
  •  
  • Consider adding a section that discusses potential off-target effects of M-NVs in non-tumor tissues. Are there concerns about unwanted cytotoxicity in healthy brain cells?

R: To join altogether the requested from both the Referees (the former asking viability data related to the RB loaded M-NVs and the latter on cytotoxicity in non tumor brain cells), we have performed an additional experiment considering MTT viability assays in normal rat astrocytes vs U138-MG human glioma cells, assaying empty M-NVs (using identical conditions reported in the manuscript) and RB-loaded M-NVs (12.5 mM), and evaluating viability trends at 24-48 and 72 hours p.t. As reported in the graph (Figure S5 panel D) a viability reduction of 50% was scored after 72 hours p.t. in glioma cells, while normal rat astrocytes, treated with empty M-NVs or with RB-loaded counterparts, did not show cytotoxicity effects along the investigated time interval

  •  
  • Given the focus on glioblastoma, have the authors considered testing M-NVs in more aggressive or resistant glioma subtypes?

R: We think this might be a next valuable suggestion, that presently in the submitted manuscript, was not primarily considered since our main focus was to firstly identify the most biologically suitable approach to generate M-NVs. Therefore, in a future perspective we might try to define homotypic or heterotypic membranes-derived M-NVs, loaded with single or combinations of compounds, to effectively induce cytotoxicity toward more aggressive or TMZ-resistant glioma subtypes.

  • How scalable is the production method for M-NVs, and what are the next steps to move this from a lab-based study to clinical trials?

R: This is another relevant topic to be considered for future experiments, particularly, before a clinical application, onto in vivo glioma induced animal models for which a significant upper scalability in the production of M-NVs will be absolutely required. Luckily, biological temples to generate M-MNs, i.e. plasma membranes, can be relatively easily isolated even in large quantities from cell culturing.

Finally, the conclusions summarizing the findings that M-NVs show promise as drug delivery systems for glioblastoma treatment. However, the long-term therapeutic efficacy, especially in in vivo models, remains to be demonstrated.

Suggestions/Improvements/questions:

  • The authors should briefly outline the limitations of their current study in the conclusion and propose future directions.
  • R: This important suggestion has been included at the end of the Discussion section.
  •  
  • Have the authors considered investigating the pharmacokinetics of M-NVs in vivo?
  • R: This point was also raised by the other Referee. This, I hope, is the reasonably justification/answer that we can provide:

“Honestly, important but also complex request additional pharmacokinetics studies, as loading efficiency, stability and drug release profiles were not performed in this contribution, but these will be for sure included in a consequent continuation of the study.

Also, hoping the Referee will understand, the Journal imposed a very limited and non-extendable time frame (not over 2 weeks !) for providing responses to the Referees, realistically not feasible to perform these studies.”

  • Are there any plans to explore potential combination therapies using M-NVs (e.g., with standard chemotherapy agents)?
  • R: Relating to this important suggestion, we have produced preliminary results of the effective co-loading of temozolomide combined with a fluorescent molecule (Berberine or the described Rose Bengal), with therefore the opportunity to track internalization by fluorescence emissions and to exploit TMZ induced toxicity

Overall recommendation: Minor revisions. The study is innovative and offers a significant contribution to the field, but addressing the above points will improve its relevance.

Round 2

Reviewer 1 Report

Comments and Suggestions for Authors

Although some concerns are not addressed, the authors explained that the studies will be done in the future.

Author Response

Dear Editor

I've replyed as round 2 to the Editor's detailed suggestions, changing the title of the manuscript and uploaded the revised version of the manuscript

Sergio Comincini
